# A Pseudorandom Number Generator Based on the Chaotic Map and Quantum Random Walks

**DOI:** 10.3390/e25010166

**Published:** 2023-01-13

**Authors:** Wenbo Zhao, Zhenhai Chang, Caochuan Ma, Zhuozhuo Shen

**Affiliations:** 1School of Electronic Information and Electrical Engineering, Tianshui Normal University, Tianshui 741000, China; 2School of Mathematics and Statistics, Tianshui Normal University, Tianshui 741000, China

**Keywords:** Li–Yorke chaos, perturbation algorithm, composition of two systems, PRNG

## Abstract

In this paper, a surjective mapping that satisfies the Li–Yorke chaos in the unit area is constructed and a perturbation algorithm (disturbing its parameters and inputs through another high-dimensional chaos) is proposed to enhance the randomness of the constructed chaotic system and expand its key space. An algorithm for the composition of two systems (combining sequence based on quantum random walks with chaotic system’s outputs) is designed to improve the distribution of the system outputs and a compound chaotic system is ultimately obtained. The new compound chaotic system is evaluated using some test methods such as time series complexity, autocorrelation and distribution of output frequency. The test results showed that the new system has complex dynamic behavior such as high randomicity, unpredictability and uniform output distribution. Then, a new scheme for generating pseudorandom numbers is presented utilizing the composite chaotic system. The proposed pseudorandom number generator (PRNG) is evaluated using a series test suites such as NIST sp 800-22 soft and other tools or methods. The results of tests are promising, as the proposed PRNG passed all these tests. Thus, the proposed PRNG can be used in the information security field.

## 1. Introduction

Chaos theory is a conspicuous area in the researches of mathematics and dynamic system and has attracted many researchers for nearly fifty years [1]. A chaotic dynamic system has the special nonlinear dynamics characteristics that can be regarded as a random motion, and its motion trail is characterized by sensitivity of the initial value and the initial parameter, unpredictability and ergodicity. Therefore, chaos theory is comprehensively applied in engineering fields of the communication, signal processing, etc. [1,2,3,4]. Especially in the information security field, many designs of safety algorithms based on the chaotic map are proposed, such as the block cipher S-box, the key generator in stream cipher, and the construction of Hash compression function, etc.

One of the most significant components of an information security system is the random number generator. Random number generator (RNG) are widely applied in many fields such as Artificial intelligence, Digital communications, System testing, Statistical simulation, Software development and Crypto-system [5,6,7,8]. In different application fields, RNG has diverse properties and these properties include: a sequence generated by a RNG has any weakness in statistics; attackers can not predict the leading sequence or the following sequence; a sequence can be generated or predicted as the internal state value is known. In view of the forgoing premises, random number generators are divided into two categories: true random number generator (TRNG) and pseudorandom number generator (PRNG). True random number generators are usually based on the phenomena of the true world and the physical process. However, TRNG has some disadvantages, such as slow speed, high cost and over dependence on hardware. Accordingly, most practical application systems choose the pseudorandom number generator, especially in network information security system (cryptographical system). A Cryptographical system requires that a PRNG with little wasted memory can generate a sequence of long period and generate unpredictable data quickly. A complex cryptosystem possesses two main operations: Diffusion and Confusion. Chaotic system has many characteristics: ergodicity, sensitivity to initial conditions and structural complexity of dynamic system. These properties are equivalent to the confusion, diffusion and algorithm complexity in traditional cryptosystem. Therefore, many Chaotic-maps-based PRNGs have been put forward. Chaotic maps such as logistic mapping and its variant, quantum logistic map, one dimension piecewise linear map and tinkerbell map have been widely used in PRNGs [1,9,10,11,12].

Most application depend on the performance of the original chaotic system, that is to say, chaos in ideal state. However, in the practical system operation, original chaos system may lead to arise problems such as short cycle, nonergodicity and decreased complexity, which will make application systems lose their original characteristics like long-term unpredictability, etc.; thus, a cryptosystem based on the original chaotic map may be successfully attacked [4,13,14]. Security of the analyzed PRNG is much lower than expected and it should be used with caution [14]. Even some security problems can allow attackers to completely crack and analyze the cryptosystems, getting the secret data and secret keys. It is critically necessary to improve chaos power performance degeneration and further optimize the chaos. Common methods of improving chaos power performance degeneration include [4]: high precision, approaches of the connection of multiple chaos systems, and methods of the disturbance, etc. Among those, the method of the disturbance can improve the performance (prolong the cycle and enhance the complexity) of the chaos greatly if constructed rationally. We usually hope to get the chaos map with uniform output; it is necessary to further optimize the output distribution. A brilliant simple solution to optimize the output distribution can be chosen, which is to combine chaotic outputs with another pseudorandom signals.

On the basis of the fact that ring graph quantum random walks (QRWs) are prone to generate the pseudorandom sequence with uniform distribution, the system output distribution can be improved by mixing original system outputs and QRWs outputs together. QRWs is a quantum corresponding scene of the classical random walk. For the widespread applications of the classical random walk in fields of physics, biology, computer science and finance, etc. [15]. Hence in the future, QRWs probably become tools for many applications, and it may appear lots of information security algorithms based on the QRWs [16,17,18,19]. In literature [18], Y. Yang and Q. Zhao constructed a novel PRNG based on QRWs. The present QRWs-based PRNG has some advantages such as better statistical complexity and recurrence, whose normalized Shannon entropy are close to 1. Thus, it is indicated that outputs of PRNG based on QRWs distribute uniformly. Therefore, it is a good method by simulating QRWs to construct a “stochastic” system.

In conclusion, if a PRNG based on the chaotic map is to be designed, the original chaotic system is should not be used directly. A perturbation algorithm (makeing use of a high dimensional chaotic system to disturb the inputs and parameters of the original system) should be applied to enhance randomness of chaotic system and expand its key space. The devise of combining the chaotic system with a sequence based on QRWs will be further improved output distribution. A predicted outcome is that a compound chaotic system with large key space, high randomness and high uniform output distribution can be obtained.

Inspired by reasons discussed above, we are motivated to search for a novel compound chaotic system with complex dynamic behavior and design a PRNG based on compound chaotic system to meet the needs of practical applications.

The rest of this paper is arranged as following: In Section 2, we constructed a surjective chaotic map that satisfies Li–Yorke chaos condition in unit region; in Section 3, we used a discrete two-dimensional chaotic system to disturb the parameters and inputs of the constructed system and combined its outputs with the sequence generated by the quantum random walk, thus obtaining a compound chaotic system with complex behavior and nearly uniform distribution; in Section 4, a new scheme for generating pseudorandom numbers is presented utilizing the composite chaotic system, and the security and randomness of the proposed PRNG are analyzed and tested roundly; in Section 5, the research results are summarized.

## 2. A Internal Randomness System Is Constructed in Unit Region

The parameter equations of conic curve in unit region are given:(1)$$\left\{\begin{array}{c} x\left(t\right)=\frac{2\;\omega_{1}\;t\left(1-t\right)x_{1}+\omega_{2}\;t^{2}}{\omega_{0}\;{\left(1-t\right)}^{2}+2\;\omega_{1}\;t\left(1-t\right)+\omega_{2}\;t^{2}},\\ y\left(t\right)=2\;\frac{\omega_{1}\;t\left(1-t\right)y_{1}}{\omega_{0}\;{\left(1-t\right)}^{2}+2\;\omega_{1}\;t\left(1-t\right)+\omega_{2}\;t^{2}}. \end{array}\right.$$When t∈[0,1], the two ends of the curve are $\left(0,0\right)$ and $\left(1,0\right)$. The shape of the curve is determined by $\omega_{0}$, $\omega_{1}$ and $\omega_{2}$, and the bump and height of the curve are determined by $\left(x_{1},y_{1}\right)$. The curve is shown in Figure 1(1).

As shown in Figure 1(1): point A is the maximum value of the curve, the ordinate of point A is the same as the abscissa of point C, the abscissa of point A is the same as the abscissa of point E, and the abscissa of point C is the same as the abscissa of point D. Point A is mapped to point D through two times of recursion. According to the conclusion in literature [20], map (Equation 1) satisfied the general conditions for Li–Yorke chaos as long as the ordinate of point D is greater than the ordinate of point E. The system satisfying conditions for Li–Yorke chaos should be constructed from the explicit form of the curve because explicit form is more understandable than implicit expression. There are three types of conic curves, namely, parabola, ellipse and hyperbola. In this paper, we choose an ellipse curve for researching and focus on constructing a chaotic system.

Let *f* be an elliptic curve. Because the two endpoints of the curve are $\left(0,0\right)$ and $\left(1,0\right)$ respectively, the curve equation can be given:(2)$$f\left(x\right)=a\sqrt{-x^{3}+x},$$
calculate the maximum value of curve $D\left(x_{max},y_{max}\right)$, we have
$$\left\{\begin{array}{c} x_{max}=\frac{\sqrt{3}}{3},\\ y_{max}=f\left(x_{max}\right). \end{array}\right.$$
If Equation (Equation 2) satisfied the following conditions (Equation 3):(3)$$\left\{\begin{array}{c} 1\geq y_{max}>0,\\ f^{2}\left(y_{max}\right)>x_{max}, \end{array}\right.$$
system (Equation 2) is Li-Yorke chaos as (1.4690<a≤1.6110) by calculation.

Chaotic map satisfying conditions (Equation 3) is transformed to surjective map by isometric scaling, and it does not change chaotic characteristics. So, curve (Equation 2) is first shifted to the left and down $m=f(y_{max})$, as shown in Figure 1(2). Then, the map is magnified by $t=1\;/\;\left(y_{max}-f(y_{max}\right)$ times and obtains a surjective chaotic map in unit region. The surjective map is shown in Figure 1(3), and the expression is as the following:(4)$$g\left(x\right)=t\left(a\sqrt{\frac{1}{t}x+m-{\left(\frac{1}{t}x+m\right)}^{3}}-m\right).$$

### Lyapunov Exponent, Trajectory Iteration Diagram and Bifurcation Diagram

Lyapunov exponent is a main quantitative index of chaotic analysis by reason that it is used to describe the local stability of the trajectory of the dynamic system. In general, as the system is chaotic, the Lyapunov exponent is positive. The calculation of Lyapunov exponent is by using the definition method, and the evaluating expression is as the following:(5)$$\left\{\begin{array}{c} g^{n}\left(x\right)=g\left(g^{n-1}\left(x\right)\right),\\ LE=\lim\frac{1}{n}\sum_{i=0}^{n-1}\ln{\left|\frac{dg^{n}\left(x\right)}{dx}\right|}_{x=x_{i}}. \end{array}\right.$$
The Lyapunov exponent of system (Equation 4) calculated by Formula (Equation 5) is shown in Figure 2. Figure 2 shows the Lyapunov exponent of system (Equation 4) for different control parameter *a*. According to Figure 2, the system (Equation 4) can exhibit chaotic behavior for 1.52≤a≤1.6.

Bifurcation diagram of map (Equation 4) is shown in Figure 3. From Figure 3, the results indicate that as parameter *a* gradually increases from 1.5 to 1.61, chaos phenomena appears. In a certain range of values of the control parameter, 1.59≤a≤1.60, full chaotic behavior can be seen with Figure 2 and Figure 3.

For a chaotic system, it will be found that the iterative trajectory of the system will present chaotic state as giving an initial value and analyzing its output sequence. The iterative trajectory of system (Equation 4) is shown in Figure 4. It can be judged from the Figure 4 that as a=1.59, the system has obvious chaos characteristics; as a=1.44, the system takes on periodic oscillation state; as a=1.38, the system converges to a stable state.

To facilitate the simplification of the established chaotic system, let $\alpha =\left(a-1.56\right)\times 20+1$, then Equation (Equation 4) can be rewritten as:(6)$$\begin{array}{cc} g\left(x\right)=t\left(a\left(\alpha \right)\sqrt{\frac{1}{t}x+m-{\left(\frac{1}{t}x+m\right)}^{3}}-m\right), & \;a\left(\alpha \right)=\frac{\alpha -1}{20}+1.56. \end{array}$$

## 3. Design and Performance Analysis of a New Compound Chaotic System

In practical applications, a chaotic system with large key space, complex dynamic behaviors and nearly uniform distribution is generally required. As a chaotic system has been in operation of digital systems with finite precision, the dynamic performance can deteriorate. So, to impove morely the dynamic performance of the chaotic system (Equation 4) and overcome the short period of chaotic sequence caused by the finite precision effect, a mechanism needs to be designed.

Firstly, we choose perturbation method [4,21], that is, known two-dimensional chaotic map outputs are used to perturb the constructed system parameters and inputs; Then, we research the quantum random walk on the ring graph under control of two-dimensional chaos, and the outputs of the perturbed system are merged into outputs of the quantum random walk. A new compound chaotic system with complex behavior is obtained and the output sequence generated by new system distributes uniformly in the whole state space.

### 3.1. Optimization Algorithm Based on Two-Dimensional Chaotic Map and Quantum
Random Walk

The discretization of (Equation 6) is derived as the following:(7)$$x_{n+1}=t\left(a\left(\alpha \right)\sqrt{\frac{1}{t}x_{n}+m-{\left(\frac{1}{t}x_{n}+m\right)}^{3}}-m\right).$$Call equation (Equation 7) as the Ecsys; then, we select a two-dimensional hyperchaotic system to disturb the parameters and inputs of Ecsys while controlling the quantum random walk.

#### 3.1.1. Two-Dimensional Hyper-Chaotic System

The general expression of two-dimensional hyperchaotic system is as the following:(8)$$\begin{array}{cc} u_{n+1} & =k_{11}+k_{12}u_{n}+k_{13}u_{n}^{2}+k_{14}v_{n}+k_{15}v_{n}^{2}+k_{16}u_{n}v_{n},\\ v_{n+1} & =k_{21}+k_{22}u_{n}+k_{23}u_{n}^{2}+k_{24}v_{n}+k_{25}v_{n}^{2}+k_{26}u_{n}v_{n}. \end{array}$$Limit the coefficients in Equation (Equation 8) and make most of them zero, a simplified two-dimensional chaotic system can be obtained ultimately. Let $k_{11}=k_{12}=k_{13}=k_{16}=k_{21}=k_{23}=k_{25}=k_{26}=0$, that is
(9)$$\begin{array}{c} u_{n+1}=k_{14}v_{n}+k_{15}v_{n}^{2},\\ v_{n+1}=k_{22}u_{n}+k_{24}v_{n}, \end{array}$$
where $k_{15}=-1.55$, $k_{22}=-1.1$ and $k_{24}=0.1$. We take $k_{14}$ as the control parameter.

In order to discuss chaotic characteristics of system (Equation 9) caused by the variation of parameter $k_{14}$, a modified version of Marotto’s theorem is first presented in literature [22]. A discrete dynamical system is as the following:(10)$$X_{n+1}=F\left(X_{n}\right),n\geq 0,$$
where F:X→X is the mapping, and $\left(X,∥\cdot ∥\right)$ is the Banach space.

**Theorem** **1.**
*Let $z\in R^{n}$ be a fixed point of the mapping $F:R^{n}\to R^{n}$. Assume that*
*a* *.*
*F is continuously differentiable in some fields of z and the absolute values of all eigenvalues of DF(z) are greater than 1. Thus, there exists a normal number r and a norm of $∥\cdot ∥$, so that F can expand on ${\bar{B}}_{r}\left(z\right)$ under $∥\cdot ∥$, ${\bar{B}}_{r}\left(z\right)$ is a closed sphere of space $\left(R^{n},∥\cdot ∥\right)$ centered on z;*
*b* *.*
*z is the return-expansion fixed point of F, that is, it exists a point $x_{0}\in B_{r}\left(z\right)$ and positive integer m such that $F^{m}\left(x_{0}\right)=z\left(x_{0}\neq z\right)$, where ${\bar{B}}_{r}\left(z\right)$ is the opening ball of space $\left(R^{n},∥\cdot ∥\right)$ centered on z. F is continuous and differentiable in a field of $x_{0},x_{1},\cdots ,x_{m-1}$ and $detDF\left(x_{j}\right)$$\neq 0\left(0\leq j\leq m-1\right)$, where $x_{j}=F\left(x_{j-1}\right)$.*

*Then, the system (Equation 10) is chaotic in the sense of Li–Yorke.*


The value of $k_{14}$ is discussed below for system (Equation 9), as it satisfies theorem (1). A fixed point of system (Equation 9) is $O=\left(0,0\right)$, we can define the following norm:$$∥\left(u,v\right)∥=\sqrt{u^{2}+v^{2}}.$$Let $h=(h_{1},h_{2})$, where $h_{1}\left(u,v\right)=k_{14}v-1.55v^{2}$ and $h_{2}\left(u,v\right)=-1.1u+0.1v$. It is obvious that *h* is continuously differentiable in $R^{2}$, and its Jacobian matrix is
$$Dh\left(u,v\right)=\left(\begin{array}{cc} 0 & k_{14}-3.1v\\ -1.1 & 0.1 \end{array}\right).$$For the fixed point $O=\left(0,0\right)$, we assume that the absolute values of all eigenvalues of the matrix $Dh\left(O\right)$ are greater than 1, that is, system (Equation 9) may be chaotic when $k_{14}<-1.023532631$ or $k_{14}>1.023532631$. The Lyapunov exponent of the system is shown in Figure 5(1–2), and the bifurcation diagram of the system (Equation 8) is shown in Figure 5(3–6). It can be seen that when parameter $k_{14}$ increases gradually from 0.9 to 1.47, the system gradually enters a complex chaotic state. Without losing generality, let $k_{14}=1.5$5 and analyze whether system (Equation 9) with the fixed point $O=\left(0,0\right)$ satisfies the Theorem 1.

It is shown that $∥h\left(x\right)-h\left(y\right)∥\geq 1.1∥x-y∥$ for all $x,y\in {\bar{B}}_{r}\left(O\right)$ where $r=5.556897\times 10^{-163}$. Therefore, *O* is an expansion fixed point of *h* in ${\bar{B}}_{r}\left(O\right)$. After the calculation, there is a point $x_{0}=\left(6.253681\times 10^{-163},-1.485254\times 10^{-162}\right)\neq O$, $x_{0}\in {\bar{B}}_{r}\left(O\right)$ and a positive integer m=1290 to $h^{m}\left(x_{0}\right)=O$. It can be obtained $\frac{\partial h_{1}}{\partial u},\frac{\partial h_{1}}{\partial v},\frac{\partial h_{2}}{\partial u}$ and $\frac{\partial h_{2}}{\partial v}$ are continuous in $B_{r}\left(O\right)$, because that $\frac{\partial h_{1}}{\partial u}=0$, $\frac{\partial h_{1}}{\partial v}=1.5-2.6v$, $\frac{\partial h_{2}}{\partial u}=-1.1$ and $\frac{\partial h_{2}}{\partial v}=0.1$. Then, *h* is continuously differentiable at $x_{i}$ and $\det Dh(x_{i})\neq 0$, 0≤i≤m, according to Theorem 1, the fixed point O is the return-expansion fixed point of *h*, that is, *h* is Li–Yorke chaos. It can be inferred that map (Equation 9) is chaotic in the sense of Li–Yorke as $k_{14}=1.55$.

According to the above theoretical analysis and simulation, system (Equation 9) has complex dynamic behavior when the parameter $k_{14}\in \left[1.47,1.57\right]$. It can be used as a disturbance source to Ecsys.

#### 3.1.2. Sequence Generation Algorithm Based on Quantum Random Walk

Let *G* be n-nodes and undirected graph. It is a *n*-cycle graph, that is, the degree of each node is 2. Then, the quantum random walk in *G* contains two quantum systems: Walker and Coin. Walker is an N-dimensional Hilbert space $H_{p}$, whose location of the ground state is $\left\{|i\rangle ,i\in \left\{0,1,2,\cdots ,N\right\}\right\}$. Any position of Walker can be represented as $\sum_{i}k_{i}|i\rangle$, and $\sum_{i}{\left|k_{i}\right|}^{2}=1$. Coin is a two-dimensional dimensional Hilbert space $H_{c}$, whose ground state is $\left\{|0\rangle ,|1\rangle \right\}$. Then the state of any Coin can be expressed as $a|0\rangle +b|1\rangle$, and $\left|a^{2}\right|+\left|b^{2}\right|=1$. The joint state of Walker and Coin is $H_{t}=H_{p}⊗H_{c}$, and the evolution of the joint state is accomplished by using coin operation and position movement.

1.The coin operator ${\hat{C}}_{\theta }$ is as following:
$$\hat{C_{\theta }}=\cos\theta \left|0\rangle \langle 0\right|+\sin\theta \left|0\rangle \langle 1\right|+\sin\theta \left|1\rangle \langle 0\right|-\cos\theta \left|1\rangle \langle 1\right|.$$Let the position shift operator be ${\hat{S}}_{\left(faward,back\right)}$, and the expression is as following:
$$\hat{S}=\sum_{i}\left(\left|i+farward\left(mod\;n\right)\rangle \langle i\right|⊗\left|0\rangle \langle 0\right|+\left|i-back\left(mod\;n\right)\rangle \langle i\right|⊗\left|1\rangle \langle 1\right|\right),$$
where farward means that Walker gos right steps as the state of the coin is $|0\rangle$, and back means the steps to left as $|1\rangle$. So, each step of the quantum random walk can be written as
(11)$${\hat{U}}_{\left(\theta ,farward,back\right)}=\hat{S}\cdot \left(\hat{C}⊗I\right).$$Assuming that the initial state of the system is $|\varphi \left(0\right)\rangle$, and after *t* steps, according to Equation (Equation 11), the joint state is
(12)$$|\varphi \left(t\right)\rangle ={\hat{U}}^{t}|\varphi \left(0\right)\rangle ={\left(\hat{S}\cdot \left(\hat{C}⊗I\right)\right)}^{t}|\varphi \left(0\right)\rangle .$$Then, the probability of stopping at point $|\upsilon \rangle$ in graph *G* after step *t* is
(13)$$P_{t}\left(\upsilon |\varphi \left(0\right)\right)={\left|\langle \left(\upsilon ,0\right)|\varphi \left(t\right)\rangle \right|}^{2}+{\left|\langle \left(\upsilon ,1\right)|\varphi \left(t\right)\rangle \right|}^{2},$$
and the limiting distribution π of stopping at point $|\upsilon \rangle$ is
(14)$$\pi =\lim_{T\to \infty }{\bar{P}}_{T}\left(\upsilon |\varphi \left(0\right)\right)=\lim_{T\to \infty }\frac{1}{T}\sum_{t=0}^{T-1}P_{t}\left(\upsilon |\varphi \left(0\right)\right).$$In order to design a sequence generation algorithm, the following theorem based on Theorem 3.6 and Theorem 4.1 in literature [23] is given.

**Theorem** **2.**
*Let U be a coined quantum walk on the n-cycle graph, with n odd, and with the Hadamard transform as the coin. Then the limiting distribution π is uniform over the nodes of the graph, independent of the initial state $|\varphi \left(0\right)\rangle$.*


According to Theorem 2, if quantum random walk is based on *n*-cycle graph *G*, the number of vertices in graph *G* is *n*, *n* is an odd number, and the number of iterations t is relatively large, (Equation 14) is close to uniform distribution. A vector $\left(\theta ,faward,back,n,i_{0},c_{0}\right)$ is setup, where $c_{0}$ is the initial state of coin and $i_{0}$ is the initial position of the bludger. It can be seen from (Equation 13) that there is a nonlinear map between the probability distribution $P_{t}=\left[P_{t}\left(\upsilon_{1}|\varphi \left(0\right)\right),\;P_{t}\left(\upsilon_{2}|\varphi \left(0\right)\right),\cdots P_{t}\left(\upsilon_{n}|\varphi \left(0\right)\right)\right]$ and initial state $|\varphi \left(0\right)\rangle =\left(i_{0},c_{0}\right)$. According to (Equation 12) and (Equation 13), a uniformly distributed sequence can be generated. Hence, a sequence generation Algorithm 1 with high sensitivity to initial conditionsis is proposed following:
**Algorithm** **1** Sequence generator algorithm.Input: $\left(\theta ,\;faward,\;back,\;n,\;i_{0},\;c_{0}\right)$Output: AllOutputSeq1. $\;T_{max}=2n$, AllOutputSeq=⌀;2. $\;for\;T=1:T_{max}$  $t=1,|\varphi \left(0\right)\rangle =\left(i_{0},c_{0}\right)$,  $while\left(t<=T\right)$    $|\varphi \left(t\right)\rangle =\hat{U}|\varphi \left(t-1\right)\rangle$  end;  $|\varphi \left(T\right)\rangle =\sum_{1}^{n_{1}}k_{i}\left(x_{i},0\right)+\sum_{1}^{n_{2}}h_{j}\left(y_{j},1\right)$,  $S_{1}=\left\{\left(k_{1},x_{1},0\right),\left(k_{2},x_{2},0\right)\cdots \left(k_{n_{1}},x_{n_{1}},0\right)\right\}$,  $S_{2}=\left\{\left(h_{1},y_{1},1\right),\left(h_{2},y_{2},1\right)\cdots \left(h_{n_{2}},y_{n_{2}},1\right)\right\}$,  $RanSeq=\left\{\left(p_{1},x_{1}\right),\left(p_{2},x_{2}\right)\cdots \left(p_{n_{1}},x_{n_{1}}\right)\left|p=k^{2}\right.\right\}$  $fors_{2}=1:length\left(S_{2}\right)$    $fors_{1}=1:length\left(S_{1}\right)$      $if\left(x_{s_{1}}==y_{s_{2}}\right)$        $\left(p_{s_{1}},x_{s_{1}}\right)=$$\left(p_{s_{1}}+h_{s_{2}}^{2},x_{s_{1}}\right)$,        break,      endif    endfor    $if(s_{1}==length\left(S_{1}\right))$      $RanSeq+=\left(h_{s_{2}}^{2},y_{s_{2}}\right)$,ultimately    endif  endfor,  $OutputSeq=\left\{\pm p_{i}+\frac{x_{i}}{n}\left|i=1,2\cdots length\left(RanSeq\right)\right.\right\}$,  AllOutputSeq+=OutputSeq, endfor.3. returnAllOutputSeq

#### 3.1.3. Optimization Scheme and a Compound Chaotic System

The optimization block diagram is shown in Figure 6, where *u* of system (Equation 9) is used to disturb the input *x* of Ecsys system, and *v* is used to disturb the control parameter α. A name “ Qusys ” is given by the sequence based on Algorithm 1. $\left(u,v\right)$ and output *x* of Ecsys controled parameters $\left(\theta ,c_{0},n,i_{0}\right)$ of Qusys and other parameters of Qusys are fixed. The outputs of Ecsys merged with outputs of Qusys, and $y_{j}$ is combined output. So, a new compound chaotic system is ultimately obtained.

Detailed description of perturbation algorithm is as following. A simple normalized processing function is given by
$$Fnor\left(x\right)=\left(x-x_{min}\right)\;/\;\left(x_{max}-x_{min}\right).$$
Apparently, $Fnor\left(x\right)$ can adjust the value of the output $\left(u,v\right)$ of system (Equation 9) to the unit region $\left(0,1\right)$. Two disturbance functions are constructed, and named, respectively, $T_{1}$ and $T_{2}$. The functions is designed as the following:$$\left\{\begin{array}{c} T_{1}\left(u_{i+1},x_{n}\right)=Fnor\left(u_{i+1}\right)*\beta +x_{n}*\left(1-\beta \right),\\ T_{2}\left(v_{i+1}\right)=1.5+Fnor\left(v_{i+1}\right)*\gamma , \end{array}\right.$$
where $\beta \in \left(0,1\right)$ and $\gamma \in \left(0,0.5\right)$ are two control parameters. After being perturbed, a chaotic system can be obtained:(15)$$x_{n+1}=t\left(a\left(T_{2}\left(v_{i+1}\right)\right)\sqrt{\frac{1}{t}T_{1}\left(u_{i+1},x_{n}\right)+m-{\left(\frac{1}{t}T_{1}\left(u_{i+1},x_{n}\right)+m\right)}^{3}}-m\right),$$
where $a\left(c\left(v_{i+1}\right)\right)=\frac{T_{2}\left(v_{i+1}\right)-1}{20}+1.56$. Parameter settings of Qusys are as following:$$\begin{array}{cc} \theta & =Fnor\left(u\right)\cdot \frac{\pi }{3},\\ n & =INT\left(Fnor\left(v\right)\cdot 20\right)+11,\\ i_{0} & =INT\left(x\cdot 20\right)\left(mod\;n\right),\\ faward & =INT\left(Fnor\left(v\right)\cdot 20\right)\left(mod\;n\right),\\ back & =INT\left(Fnor\left(u\right)\cdot 20\right)\left(mod\;n\right),\\ c_{0} & =1, \end{array}$$
where $INT\left(\cdot \right)$ is an Integral function. Sequence $\left\{x_{j}\right\}$ merged with sequence $\left\{q_{j}\right\}$ in a nonlinear way, and the ultimate output is
(16)$$y_{j}=\eta \left|\cos\left(j\right)\right|x_{j}+\left(1-\eta \left|\cos\left(j\right)\right|\right)q_{j},$$
where η is a proportion parameter and $\eta \in \left(0,0.2\right)$.

#### 3.1.4. The Digital Compound Chaotic System Expression

If a realized chaotic system executes on a digital device and precision of the digital device is *S* bit, a quantization function $B_{S}\left(x\right)$ is defined to analyze dynamical behavior in digital chaotic systems. Each process of specific calculation shall be quantized; for instance, y=x+z is quantified to an expression of the following form:$$y=B_{S}\left(B_{S}\left(x\right)+B_{S}\left(z\right)\right).$$In order to present a digital chaotic system expression, we have left out some details and the expression of system (Equation 6) is:(17)$$x_{n+1}=B_{S}\left(t\left(a\left(\alpha \right)\sqrt{\frac{1}{t}x_{n}+m-{\left(\frac{1}{t}x_{n}+m\right)}^{3}}-m\right)\right).$$The expression of the disturbed system (Equation 15) is as following:(18)$$\begin{array}{cc} x_{n+1} & =B_{S}(t(a\left(B_{S}\left(T_{2}\left(v_{i+1}\right)\right)\right)\\ & \;\;\cdot \sqrt{\frac{1}{t}B_{S}\left(T_{1}\left(u_{i+1},x_{n}\right)\right)+m-{\left(\frac{1}{t}B_{S}\left(T_{1}\left(u_{i+1},x_{n}\right)\right)+m\right)}^{3}}-m)). \end{array}$$According to Equation (Equation 16), the digital compound chaotic system is as following:(19)$$y_{n+1}=B_{S}\left(\eta \left|\cos\left(n+1\right)\right|\right)x_{n+1}+B_{S}\left(1-\eta \left|\cos\left(n+1\right)\right|\right)B_{S}\left(q_{n+1}\right).$$

### 3.2. Performance Evaluation of the Compound Chaotic System

#### 3.2.1. Analysis under Finite Precision

For performance assessment, we have implemented the original chaotic system (Equation 17) and the compound chaotic system in the simulation environment. The digital device is assumed to be an *S*-bit machine, and the quantization function $B_{S}$ is expressed as $B_{s}\left(x\right)=\left[x*2^{S}\right]\;/\;2^{S}$, where $\left[x\right]$ represents an integer less than or equal to *x*. Trajectories of different chaotic systems are shown in Figure 7. According to Figure 7(2), with finite precision 8 bits, trajectory of the digital Ecsys fall into periodic motion after several iterations. According to Figure 7(3), for the digital compound chaotic system, periodic motion do not occur as 8 bits. So, the improved system still maintains strong random characteristics and good chaotic dynamics performance.

#### 3.2.2. Time Series Complexity

Approximate Entropy is to evaluate the system complexity from sequence generated by chaotic system [24]. For a sequence, the greater the approximate entropy, the higher the complexity. In literature [24], parameters for approximate entropy calculation are recommended that: mode dimension $\left(m=2\right)$, similarity tolerance $\left(r=2\right)$. The approximate entropy values of the sequence generated by the chaotic maps are calculated and shown in Figure 8.

From Figure 8, we can see that the approximate entropy values generated by the digital compound chaotic system are the largest ones among the three maps in the cases of different precisions of the digital device. In particular, low-precision digital device does not affect the new system dynamics performance; so, there is no need for additional precision compensation technology support in practical applications.

#### 3.2.3. Histogram Analysis

Histogram is a significant feature in analysis to the sequence generated by chaotic system. For a good chaotic system for encryption algorithm, the output chaotic sequence distributes uniformly in the whole state space. For the digital chaotic system, the same initial values are setup as different finite precisions, such as 8 bits and 64 bits. Sequence with length of 2424 numbers is generated respectively, and the distribution of the sequence is statistically analyzed. The statistical results are shown in Figure 9. It can be obtained from Figure 9(1–3): The distributions of original Ecsys system and digital system are mainly concentrated in the region $\left(0.9,1.0\right)$, uneven with 64 bits precision. From Figure 8, one can see that the proposed system output distributes uniformly in the total region $\left(0,1\right)$ with low precision. So, The proposed system and digital proposed system can both resist statistical attack.

#### 3.2.4. Autocorrelation Analysis

Autocorrelation is used to measure the relation between current value and past values of the same element. For sequence generated by a chaotic system, it is measuring value between own and its own shifted. It determines the presence of any repetitive patterns of bits.

If a sequence $\left\{y_{1},y_{2},\cdots ,y_{N}\right\}$, the autocorrelation function for lag *k* is as following:(20)$$r_{k}=\frac{\frac{1}{N}\sum_{i=1}^{N-k}\left(y_{i}-\mu \right)\left(y_{i+k}-\mu \right)}{\sigma^{2}},$$
where μ, σ are the mean and the standard deviation of the sequence. The autocorrelation of sequence generated by different chaotic systems is calculated for 2500 shifts in left and is plotted in Figure 10. It can be seen from Figure 10 that autocorrelation value of the original system is in the range of $\left(-0.1,0.1\right)$, while the digital compound chaotic system with finite precision is in the range of $\left(-0.05,0.05\right)$. Therefore, it can be concluded that the proposed compond chaotic system has lower autocorrelation and better correlation analysis attacks.

## 4. Design and Performance Analysis of Pseudo Random Number Generator

It is a requirement for cryptographic applications to construct pseudorandom number generator based on chaotic system [25]. However, due to the lack of strict security analysis, the PRNG based on the original chaos often has some security vulnerabilities [14,26]. If the algorithm for PRNG is reasonably designed or the PRNG is designed based on the proposed system with more chaotic behavior, resistance against the finite precision effect, and larger key space, the PRNG should have good random characteristics, security and effectiveness.

### 4.1. Design of PRNG Based on the Proposed Compond Chaotic System

A PRNG can be used for any application if the PRNG has some properties such as good statistical properties, long cycle length, larger key space, etc. In order to achieve a fast throughput and make easier hardware or soft implementation, mechanism with *p* bit accuracy is adopted. The steps of algorithm for generating pseudo-random numbers are as following:1.Import the keys: initialize $\left(u_{0},v_{0}\right)$, $k_{14}$ and $x_{0}$, which are the control parameters and initial conditions as shown in Figure 6.2.Iterate the proposed compound chaotic map 1000 times and the output $y_{i}$ is discarded, where *i* end at 999.3.Generate and output the random number $z_{n}$ using the following equation from the $y_{n}$:
$$z_{n}=\left[y_{n}\times 10^{m}-floor\left(y_{n}\times 10^{m}\right)\right]\times 2^{p},$$
where *p* is the length of the corresponding binary random number and *n* start at 1000. The expression $y_{n}\times 10^{m}-floor\left(y_{n}\times 10^{m}\right)$ excludes m most effective numbers, which makes it more complex and uniform. The value of *m* is determined by *p*, and the proposed values are listed in Table 1.

### 4.2. Analysis and Test of Security for the Proposed PRNG

It’s essential for a random number generator to perform all necessary analyses and tests. There are several fundamental analysis and tests to verify the randomness, security and availability of proposed algorithm. The pseudorandomness of sequence generated by RNG is mainly through recurrence plots analysis, information entropy, and random evaluation software to test, etc. The following excerpt is that under p=2 and m=2 conditions, some security characteristics of the PRNG are comprehensively analyzed or tested and the test results with the fine-grained trace are carried out.

#### 4.2.1. Key Space Analysis

Random number generator is mainly used to generate key and from an encryption point of view, the size of the key should not be less than $2^{128}$ to provide a high level of security [27]. We select $\left(u_{0},v_{0},k_{14},x_{0}\right)$ as the key set. These parameters should be selected from the control parameters and initial conditions of the chaotic region, which depends on the system bifurcation diagram. Because bifurcation diagram is used to describe mutations in system dynamics. Thus, a set of keys are given based on the bifurcation diagram of system (Equation 4), including: $u_{0}\in \left[-0.9,0.3\right]$, $v_{0}\in \left[-0.4,1.2\right]$, $k_{14}\in \left[1.47,1.57\right]$ and $x_{0}\left(0.0,1.0\right)$. If the computational accuracy of the actual system applied is $10^{16}$, the range of the secret key space can be roughly calculated as following:$$\left(6\times 10^{15}\right)\left(8\times 10^{15}\right)\left(10^{15}\right)\left(10^{16}\right)=4.8\times 10^{62}≃2^{208}.$$It can be concluded that the size of the key space is sufficient to resist all kinds of violent attacks.

#### 4.2.2. Correlation Analysis

For a PRNG, the correlation coefficients between sequences produced with nearby keys are computed according to the method in [28]. For two sequences $S_{1}=\left[x_{1},x_{2},\cdots ,x_{N}\right]$ and $S_{2}=\left[y_{1},y_{2},\cdots ,y_{N}\right]$, coefficients are calculated as following:(21)$$Cor\left(S_{1},S_{2}\right)=\frac{\sum_{i=1}^{N}\left(x_{i}-\bar{x}\right)\cdot \left(y_{i}-\bar{y}\right)}{{\left[\sum_{i=1}^{N}{\left(x_{i}-\bar{x}\right)}^{2}\right]}^{1/2}{\left[\sum_{i=1}^{N}{\left(y_{i}-\bar{y}\right)}^{2}\right]}^{1/2}},$$
where $\bar{x}$ and $\bar{y}$ are the mean values of *x* and *y* respectively. Correlation is strong between two sequences for $Cor\left(S_{1},S_{2}\right)≃1$ and no or very small correlation corresponds to $Cor\left(S_{1},S_{2}\right)≃0$.

Let p=32 and m=2; the correlation coefficients test are performed as following:1.$x_{01}=0.205001025,u_{01}=0.102772828$ and $v_{01}=0.118667888$, a sequence with $10^{6}$ numbers is generated. If the initial condition changes small ($x_{02}=x_{01}+0.000000000000001$) and the others remain unchanged, a new sequence will be generated.2.Let the initial condition *u* is changed ($u_{02}=u_{01}+0.000000000000001$) and the others remain unchanged, a new sequence with the same length will be generated.3.Let the initial condition *v* is changed ($v_{02}=v_{01}+0.000000000000001$) and the others remain unchanged, a new sequence with the same length will be generated.

Correlation coefficients data are calculated by formula (Equation 21) and listed in Table 2. By analyzing the data in Table 2, the following conclusions can be drawn: there is no correlation between the generated sequences produced the proposed PRNG is sensitive to small changes in all initial conditions.

#### 4.2.3. Recurrence Plots Analysis

A powerful tool is given in [29] for visualization and analysis of recurrence, called recurrence plot (RP). As analyzing nonlinear time series by RP, phase space reconstruction is the first step. The phase space reconstruction is carried out by selecting appropriate time delay τ and embedding dimension *m*. For a time sequence $\left\{x_{n},n=1,2,\cdots ,N\right\}$, a set of *m* dimensional vectors is obtained after phase space reconstruction:(22)$$\begin{array}{c} X(n)=(x(n),x(n+\tau ),\cdots ,x(n+(m-1)\tau )),\\ n=1,2,\cdots ,N-(m-1)\tau . \end{array}$$The distance between m-dimensional vector $X\left(i\right)$ and $X\left(j\right)$ at two moments is $r_{ij}$, that is:$$r_{ij}=∥X\left(i\right)-X\left(j\right)∥.$$It can be defined as the following recursive matrix form:$$R_{ij}\left(\xi \right)=\theta \left(\xi -r_{ij}\right),i,j=1,2,\cdots ,N-\left(m-1\right)\tau ,$$
where θ is Heaviside function and ξ is threshold. Heaviside function is expressed as following:$$\theta \left(x\right)=\left\{\begin{array}{c} 1,x=\xi >r_{ij}>0,\\ 0,x=\xi ⩽r_{ij}<0. \end{array}\right.$$
For the recursive matrix, it means that the states at time *i* and time *j* are obviously different (obviously similar) when $R_{ij}=0$ ($R_{ij}=1$). The corresponding RP can be drawn according to the recursive matrix. RP can intuitively reflect the movement rule and trend of time sequences. In the actual calculation, the threshold ξ is 0.1 times of the standard deviation of the time series [30].

RP can directly show the motion rule of dynamical systems. However, RP cannot quantify the system characteristics because of the small-scale structure [31]. Recurrence quantification analysis (RQA) precisely quantifies these characteristics. RQA is proposed by Webber and Zbilut in the literature [32]. RQA is a quantitative analysis of the sequence by extracting structural feature quantities by analyzing the detailed structure of the RP. The main feature quantities are recursive rate (RR), a measure for determinism (DET), layered degrees (LAM), trapping time (TT) and the average diagonal line length (L). Small values of feature quantities for dynamical system represent a processe with weakly correlated and chaotic behaviors.

The recursion rate represents the proportion of adjacent vectors in RP and it is the percentage of recursion points in the total number of points:$$RR=\frac{1}{N^{2}}\sum_{i,j=1}^{N}R_{i,j}\left(\xi \right),$$
where *N* is the number of points on the abscissa of the RP. The histogram DLS(ξ,l) of the diagonal with length *l* in RP is as following:$$DLS\left(\xi ,l\right)=\sum_{i,j=1}^{N}\left(1-R_{i-1,j-1}\left(\xi \right)\right)\left(1-R_{i+l,j+l}\left(\xi \right)\right)\prod_{k=0}^{l-1}R_{i+k,j+k}\left(\xi \right).$$DET is the ratio of recurrence points that form diagonal structures and it is given by
$$DET=\frac{\sum_{l=l_{\min}}^{N}l\cdot DLS\left(\xi ,l\right)}{\sum_{l=1}^{N}l\cdot DLS\left(\xi ,l\right)},$$
where $l_{min}$ is the minimum length requirement (length threshold) and value is generally 2. In the RP and during *l* time steps, a diagonal line with length *L* means that a segment of system trajectory is close to another at different moments. The average diagonal length is the average time that two segments are close to each other, which can be interpreted as the average prediction time
$$L=\frac{\sum_{l=l_{\min}}^{N}l\cdot DLS\left(\xi ,l\right)}{\sum_{l=l_{\min}}^{N}DLS\left(\xi ,l\right)}.$$For calculating the number of vertical line segments VLS(v) with length *v*, a method is as following:$$VLS\left(v\right)=\sum_{i,j=1}^{N}\left(1-R_{i,j}\left(\xi \right)\right)\left(1-R_{i,j+v}\left(\xi \right)\right)\prod_{k=0}^{v-1}R_{i,j+k}\left(\xi \right).$$Laminar degree (LAM) is the ratio between the recurrence points forming the vertical structures and the entire set of recurrence points, and it can be computed,
$$LAM=\frac{\sum_{v=v_{\min}}^{N}v\cdot VLS\left(v\right)}{\sum_{v=1}^{N}v\cdot VLS\left(v\right)},$$
where $v_{\min}$ is the threshold of vertical line length (generally 2). Capture time (TT) is the average length of the vertical line segment in RP, and it is expressed as following:$$TT=\frac{\sum_{v=v_{\min}}^{N}v\cdot VLS\left(v\right)}{\sum_{v=v_{\min}}^{N}VLS\left(v\right)}.$$
Capture time (TT) is used to estimate the average duration of a system in a particular state.

Figure 11 shows RQA measures as parameter $k_{14}$ is changed, the threshold ξ=0.1σ (σ as the standard deviation of the sequence), $l_{min}=v_{min}=2$, and the initial conditions $\left(x_{0}=0.205001025,u_{0}=0.102772828,v_{0}=0.118667888\right)$. Processes with uncorrelated or weakly correlated and stochastic or chaotic behaviors cause none or short diagonals, whereas deterministic processes cause longer diagonals and less single, isolated recurrence points [18]. It can be seen from Figure 11 that RR typical measurement value is 0.5, DET value is 0.65, LAM value is 0.03 and TT value is 3. Obviously, these values are small and the proposed PRNG has good randomness.

#### 4.2.4. Information Entropy

One of the most important concepts in information theory is entropy, which is first introduced by Shannon [33]. It reflects the uncertainty and randomness of each information system. Entropy measures the unpredictability of a sequence generated by a PRNG. For a sequence $S=\left\{x_{1},x_{2},\cdots ,x_{n}\right\}$, the definition of information entropy $H\left(S\right)$ is as follows:(23)$$H\left(S\right)=\sum_{i=1}^{N}p\left(a_{i}\right)log_{2}\frac{1}{p\left(a_{i}\right)},$$
where number $a_{i}$ has the probability of $p\left(a_{i}\right)$ to occur in sequence *S*. In actual calculation, a corresponding character sequence is generated for every 8 bits by original binary sequence. The information entropy is calculated according to Formula (Equation 23). For a sequence of bytes, the ideal value of information entropy is 8.

Sequences of varying lengths and the different initial conditions are generated by the proposed PRNG, and their entropy values are shown in Table 3. The values of the different initial conditions with $k_{14}=1.55$ are: $S_{1}$ ($x_{0}=0.105001025,\;u_{0}=0.201772828,\;v_{0}=0.218667888$), $S_{2}$ ($x_{0}=0.105001025,\;u_{0}=0.201772828,\;v_{0}=0.218667888$) and $S_{3}$ ($x_{0}=0.505001025,\;u_{0}=0.251772828,\;v_{0}=0.418667088$). Sequences with lengths of 5000, 10000, 20000, 100000 and $10^{7}$ are generated. Table 3 shows that each information entropy value is close to the ideal value 8.

#### 4.2.5. Statistical Complexity Measure

Complexity is a measure of off-equilibrium ‘order’ [18]. Statistical complexity measure (SCM) is proposed as quantifiers of the degree of physical structure in a signal [34]. SCM can be used to study the complex structure hidden in chaotic system. In literature [35], the statistical complexity of the presented algorithm is calculated. The probability distribution *P* is associated with the time series generated by the dynamical system. The intensive SCM ($C_{j}\left[P\right]$) can be considered as a quantity that characterizes the probability distribution *P* which not only quantifies the randomness but also presents the structure. $C_{j}\left[P\right]$ is defined based on information entropy as following:(24)$$C_{j}\left[P\right]=Q_{j}\left[P,P_{e}\right]\cdot H_{S}\left[P\right],$$
where $H_{S}\left[P\right]$ is the entropic measure and $Q_{J}$ is “disequilibrium”. $H_{S}\left[P\right]$ and $Q_{J}$ are defined in [36]. $Q_{J}$ is given by
$$Q_{j}\left[P,P_{e}\right]=Q_{0}\cdot \left\{H\left[\left(P+P_{e}\right)/2\right]-H\left[P\right]/2-H\left[P_{e}\right]/2\right\},$$
where $Q_{0}$ is a normalization constant that reads
$$Q_{0}=-2{\left\{\left(\frac{N+1}{N}\right)\ln\left(N+1\right)-2\ln\left(2N\right)+\ln N\right\}}^{-1}.$$$H_{S}\left[P\right]$ is defined as following:(25)$$H_{S}\left[P\right]=H\left[P\right]/H_{max},$$
where $H\left(P\right)$ is the Shannon entropy. For an extremely good PRNG based on chaotic system, it can be expected that “no attractor” will be reconstructed. It will be quite reasonable to obtain a homogeneity cloud of points with a tendency to fill the d-dimensional space [35]. Consequently, the associated permutation probability distribution will be $P≃P_{e}$ [9]. So, in the case of a PRNG, the “ideal” values are $H_{S}\left[P\right]≃1$ and $C_{J}\left[P\right]≃0$.

If entropy $H_{S}$ and the intensive statistical complexity $C_{J}$ are as functions of the number of 8 bits-words, then $H_{max}=8,\;N=256$ and $P_{e}=\left\{1/N,1/N,\cdots ,1/N\right\}$. Based on the calculations mentioned above, the normalized entropy $H_{s}$ and the intensive statistical complexity $C_{j}$ as functions of the number of 8 bits are shown in Figure 12. It can be obtained from the Figure 12 that $C_{j}$ and $H_{s}$ tend to 0 and 1, respectively. So, the proposed PRNG is successfully verified by the statistical complexity and the normalized Shannon entropy.

#### 4.2.6. Degree of Non-Periodicity

Wavelet analysis is a valuable tool for the study of dynamic systems. The Scale index technique and the windowed Scale index are based on the continuous wavelet transform and the wavelet multi-resolution analysis [37,38]. The tools are designed to measure the degree of non-periodicity through its wavelet scalogram, allowing to quantify how much chaotic a signal is [38].

In order to detect and study nonperiodicity in sequences generated by PRNG, we can regard the PRNG as a continuous function $f\in L^{2}\left(R\right)$, where *f* defines the time interval at a finite time interval $I=\left[a,\;b\right]$ and *I* is large enough [18]. The Continuous Wavelet Transform (CWT) of *f* at time *u* and scale *s* is defined as following:$$Wf\left(u,s\right):=\langle f,\psi_{u,s}\rangle =\int_{-\infty }^{+\infty }f\left(t\right)\psi_{u,s}^{*}\left(t\right)dt,$$
and it provides the frequency details of the function corresponding to scale *s* and time location *u*. The scalogram of *f* at a given scale *s* is given by
$$S\left(s\right)=∥Wf(u,s)∥={\left(\int_{-\infty }^{+\infty }{\left|Wf(u,s)\right|}^{2}du\right)}^{\frac{1}{2}}.$$
$S\left(s\right)$ is the energy of the CWT of *f* at scale *s*. The scalogram is a useful tool for studying a signal because it can detect the most representative scales or frequencies. The innerscalogram of *f* at a scale *s* can be defined by:$$S^{inner}\left(s\right)={∥Wf(u,s)∥}_{J\left(s\right)}={\left(\int_{c\left(s\right)}^{d\left(s\right)}{\left|Wf(u,s)\right|}^{2}du\right)}^{\frac{1}{2}},$$
where $J\left(s\right)=\left[c\left(s\right),\;d\left(s\right)\right]\subseteq I$ is the maximal subinterval in *I* for which the support of $\psi_{u,s}$ is included in *I* for all $u\in J\left(s\right)$. Let *l* be the length of $\psi_{u,s}$ and b−a≫sl must also be satisfied. Since the length of $J\left(s\right)$ depends on the scale *s*, the values of the inner scalogram do not be compared at different scales. In order to avoid this problem, the inner scalogram should be normalized as follows:$${\bar{S}}^{inner}\left(s\right)=\frac{S^{inner}\left(s\right)}{{\left(d\left(s\right)-c\left(s\right)\right)}^{\frac{1}{2}}}.$$
In [38], the new Scale index of *f* in the scale interval $\left[s_{0},\;s_{1}\right]\subseteq I$ is given by the quotient
$$i_{scale}=\frac{S^{inner}\left(s_{min}\right)}{S^{inner}\left(s_{max}\right)},$$
where $s_{max}\in \left[s_{0},\;s_{1}\right]$ is the maximal scale such that $S^{inner}\left(s_{max}\right)\geq S^{inner}\left(s\right)$ for all $s\in \left[s_{0},\;s_{1}\right]$, and $s_{min}\in \left[s_{max},\;2s_{1}\right]$ is the smallest scale such that $S^{inner}\left(s_{min}\right)\leq S^{inner}\left(s\right)$ for all $s\in \left[s_{max},\;2s_{1}\right]$. From its definition, the scale index $i_{scale}$ is such that $0\leq i_{scale}\leq 1$, and it can be interpreted as a measure of the degree of nonperiodicity of the signal [37,38].

Let haar wavelet be mother wavelet function to calculate the Scale index $i_{scale}$, where Δs=Δt=0.05s, $s_{0}=1$ and $s_{1}=20$. Figure 13 shows the Scale index analysis of the proposed PRNG, Henon map and the logistic map. It is apparent from comparison of Figure 12 and Figure 13, that the Scale index of the proposed PRNG is higher than other two chaotic maps. Thus, the generated sequence of the proposed PRNG is highly nonperiodic.

The windowed Scale index is appropriate for nonstationary time series whose characteristics change over time [38]. It is based on the windowed scalogram and the scale index. The windowed scalogram of a time series *f* is given by [39]:$$WS_{\tau }\left(t,s\right)={\left(\int_{t-\tau }^{t+\tau }{\left|Wf(u,s)\right|}^{2}du\right)}^{\frac{1}{2}},$$
where *f* is centered at time t with radius τ. The windowed scale index of a time series *f* centered is defined as
$$wi_{scale,\;\tau }\left(t\right)=\frac{WS_{\tau }\left(t,s_{min}\right)}{WS_{\tau }\left(t,s_{max}\right)},$$
where $s_{max}$ is the smallest scale such that $WS_{\tau }\left(t,s_{max}\right)\geq WS_{\tau }\left(t,s\right)$ for all $s\in \left[s_{0},\;s_{1}\right]$, and $s_{min}$ is the smallest scale such that $WS_{\tau }\left(t,s_{min}\right)\leq WS_{\tau }\left(t,s\right)$ for all $s\in \left[s_{max},\;2s_{1}\right]$.

In general, if $\left[a,b\right]$ is the support of *f*, $\tau =\left(b-a\right)/20$ is a good choice [38]. Figure 14 shows the windowed scale index analysis of the proposed PRNG ($k_{14}=1.55$), Henon map (b=0.3, a=1.155) and the logistic map (a=3.88). As can be seen in Figure 14, the windowed Scale index clearly shows the evolution over time. Through comparative analysis, it can be obtained that window scale index of the sequence generated by the proposed PRNG changes over time mainly above 0.75, which is much higher than the other two classical chaos maps. Thus, the generated sequence of the proposed PRNG is highly nonperiodic over time.

#### 4.2.7. Differential Attack

Differential attack is that the effect of corresponding ciphertexts is analyzed as small changes on the plaintext. For a PRNG, it is applied the same analysis on the initial seeds which are at the same time keys because there is no plaintext. “Bit Change Rate (BCR)” is carried out to ensure the resistance of the proposed PRNG against the differential attack.

Bit Change Rate (BCR) criterion is defined as
(26)$$BCR\left(S_{1},S_{2}\right)=\frac{Bit_{Diff}\left[S_{1},S_{2}\right]}{N}\times 100\%,$$
where $S_{1}$($S_{2}$) is the sequence generated on the initial seed “$seed_{1}$” (“$seed_{2}$” ) and *N* is the generated sequence length and $∥seed_{1}-seed_{2}∥<\varepsilon$, which represents the small change between seed keys. $Bit_{Diff}\left[S_{1},S_{2}\right]$ is the number of different bits in $S_{1}$ and $S_{2}$. If the measure of BCR for the two sequences is close to 50, it indicates the two sequences are almost completely different. The results of BCR for sequences with $10^{7}$ bit lengths, is displayed with small changes ($10^{-15}$) on initial conditions in Table 4. From Table 4, it can be clearly seen that the BCR values are close to 50 percent. Hence, the proposed PRNG is sensitive to the change of seeds, and it can be concluded that the presented PRNG is highly resistive against differential attack.

#### 4.2.8. Random Tests

To examine the randomness of sequence generated by presented PRNG, NIST SP800-22 is carried out. The soft test suit includes 17 independent statistical tests, which focus on a sort of different types of nonrandomness in sequence. This software mainly uses performance indicator *p*-value which determined the random performance of the sequence. If the *p*-value of sequence is higher than the threshold α (the significance level), it means that the sequences pass the test. In our tests, a bit sequence is generated, which had the length of $100\times 10^{6}$ bits and the bit sequence was divided into 100 subsequences. α is 0.01, which implies that the sequence can be inferred to be random with 99% probability if it passes the test.

By this way, results from all statistical tests are given in Table 5. From Table 5, the results of sequence generated by proposed PRNG are all “success”. Hence, the proposed PRNG successfully passed the NIST SP800-22 tests.

#### 4.2.9. Speed Performance Analysis

Speed of data generated by PRNG is an important factor for evaluating the performance. For the proposed PRNG, the time cost is measured in the running environment: Centos 7, Intel I5-6300U CPU, 4GB RAM and MATLAB 2018a software framework. We measured the time cost in the running environment. We set the parameters and 100 bit sequence are generated, each of which is 500,000 bits in length. Table 6 compares the speed of the proposed PRNG in terms of the number of bits generated with other PRNG schemes. Compared with other PRNG schemes, the proposed PRNG is fast enough for practical application.

## 5. Conclusions and Discussion

In a summary, we proposed a new scheme with perturbation and mixture together to optimize the one-dimensional chaotic map self-constructed. We obtain a compound chaotic system using this scheme and the new system is evaluated using some test methods such as time series complexity, autocorrelation and distribution of output frequency. The test results showed that the new system has high randomicity and the system can operate well in the environment of low precision equipment. Thus, it can be concluded that the new scheme can be used to design a compound chaotic system with complex dynamic behavior. When using the compound chaotic system in a PRNG, designer can easily achieve high security and good quality of the random bit sequences.

A pseudorandom number generator (PRNG) is specifically designed based on the new chaos. The PRNG merely relies on the equations used in compound chaotic map. The algorithm is not complex, which does not impose high hard-ware requirement and thus speed is fast. The proposed PRNG exhibits excellent security property in terms of quantifiers based on information theory, recur rence plots, nonperiodity, correlation anaysis, differential attack and NIST tests. By these test results, we conclude that our PRNG is a reliable PRNG and it can generate highly available random numbers for various applications in computer science.

## Figures and Tables

**Figure 1 entropy-25-00166-f001:**
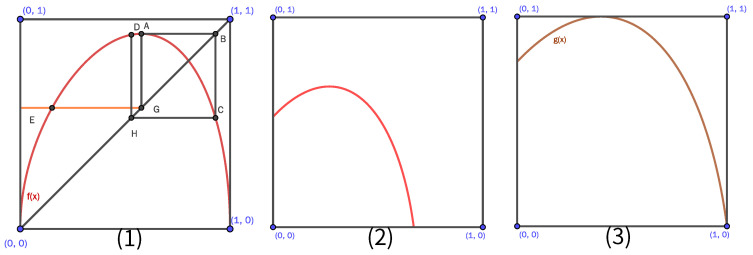
Conical section in unit region.

**Figure 2 entropy-25-00166-f002:**
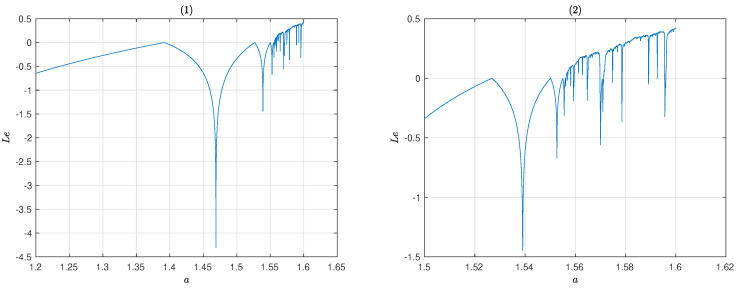
Lyapunov exponent of the map (Equation 4).

**Figure 3 entropy-25-00166-f003:**
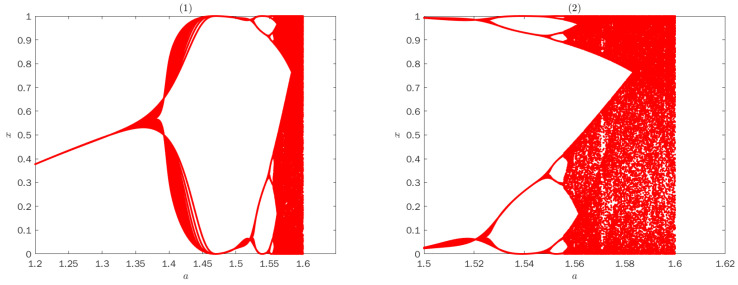
Bifurcation diagram of the map (Equation 4).

**Figure 4 entropy-25-00166-f004:**
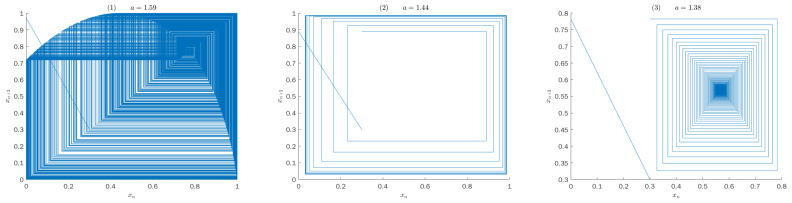
Iterative trajectory of map (Equation 4).

**Figure 5 entropy-25-00166-f005:**
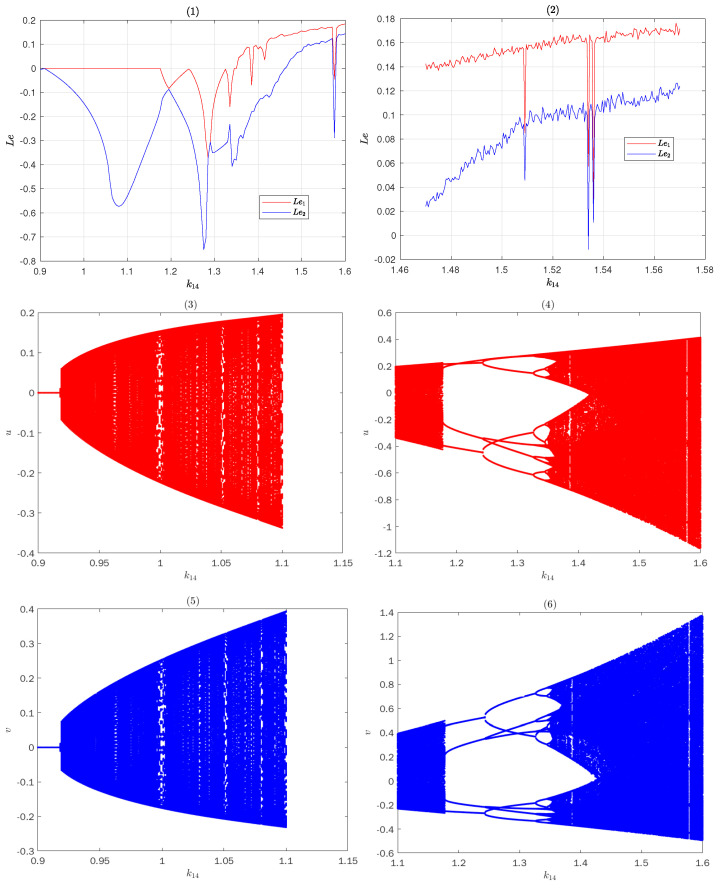
Lyapunov exponent and Bifurcuresation diagram of sys (Equation 9) for $k_{14}$.

**Figure 6 entropy-25-00166-f006:**
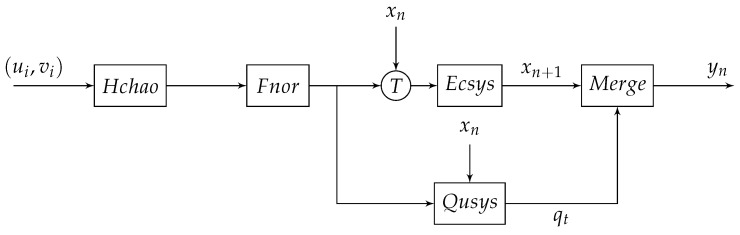
Block diagram of the optimization scheme.

**Figure 7 entropy-25-00166-f007:**
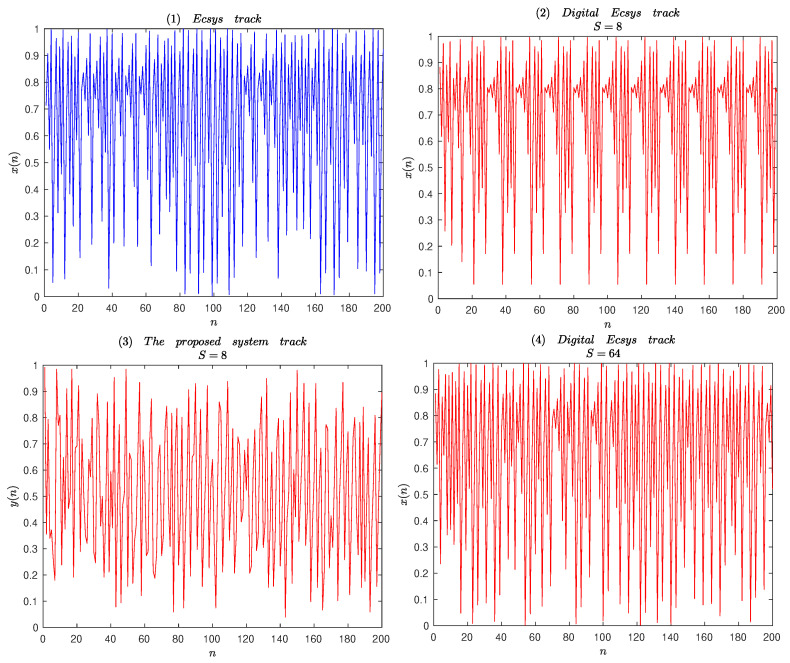
Trajectories of different chaotic systems.

**Figure 8 entropy-25-00166-f008:**
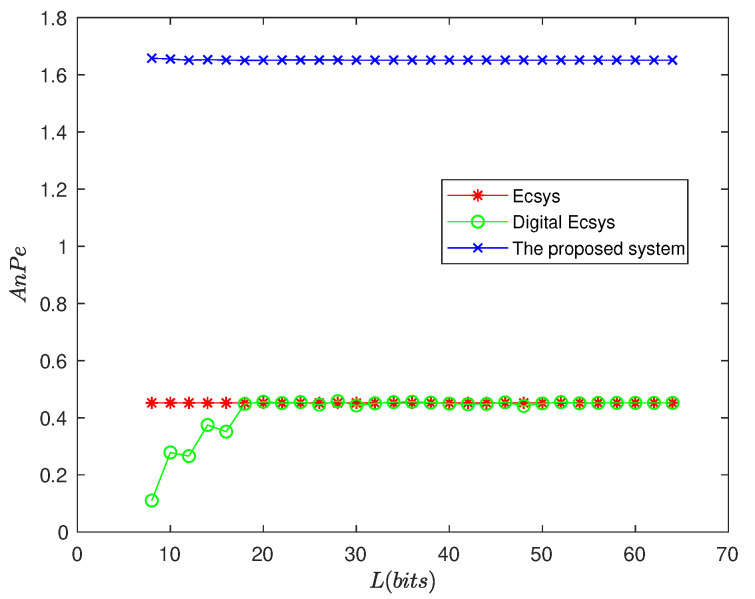
Approximate entropy.

**Figure 9 entropy-25-00166-f009:**
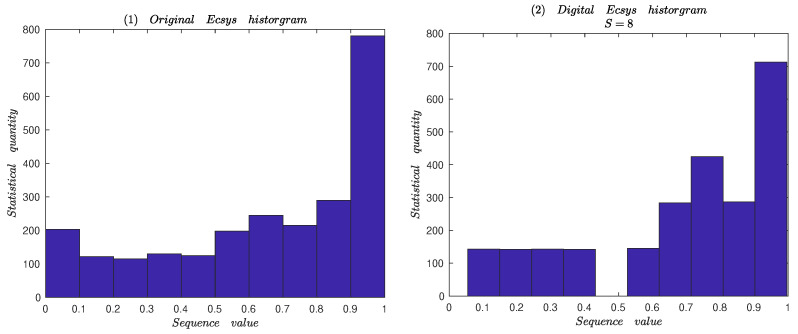
Histograms of the sequences generated by the digital chaotic system.

**Figure 10 entropy-25-00166-f010:**
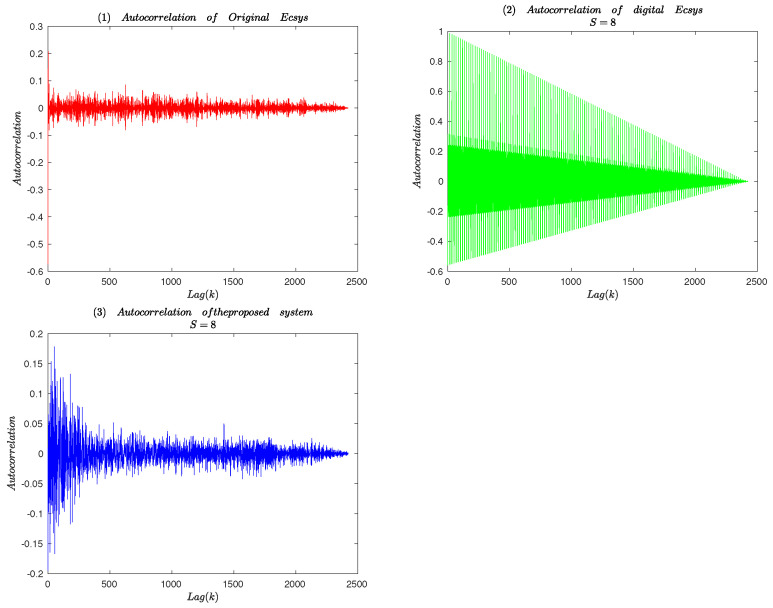
Autocorrelation analysis.

**Figure 11 entropy-25-00166-f011:**
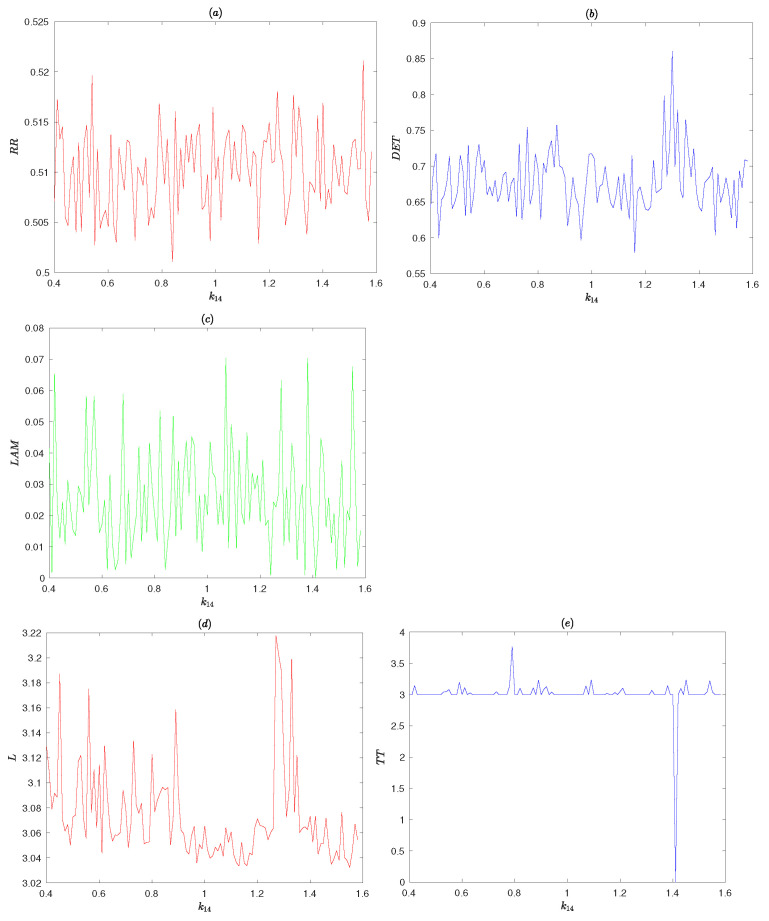
RQA measures for the proposed PRNG.

**Figure 12 entropy-25-00166-f012:**
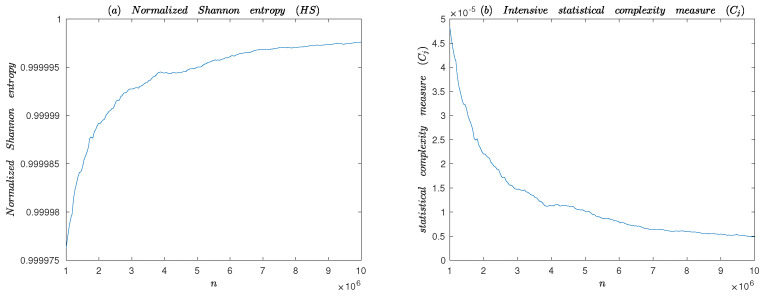
Normalized Shannon entropy $H_{s}$ and intensive SCM $C_{j}$ for the proposed PRNG.

**Figure 13 entropy-25-00166-f013:**
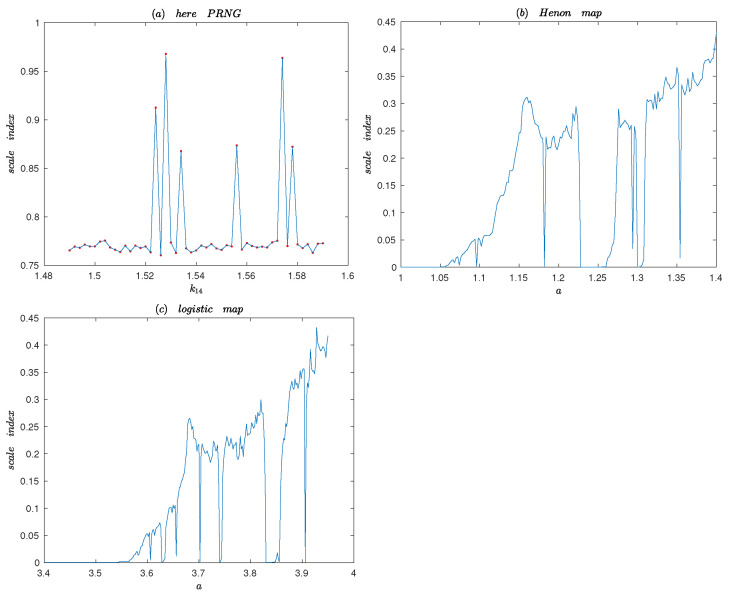
The Scale index of the Henon map, the logistic map and the proposed PRNG.

**Figure 14 entropy-25-00166-f014:**
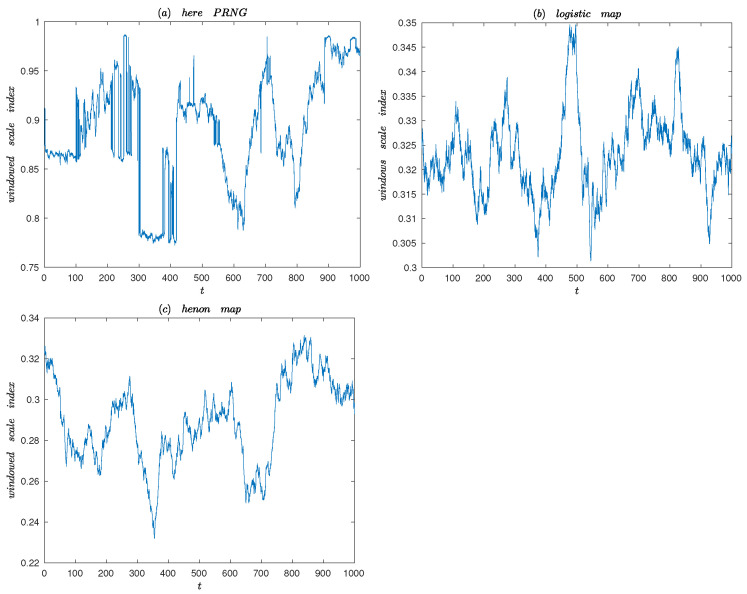
Windowed Scale index of the Henon map, the logistic map and the proposed PRNG.

**Table 1 entropy-25-00166-t001:** The suggested values of *p* and *m*.

*p*	1	4	8	16	32
*m*	≥3	≥2	≥2	≥2	≥1

**Table 2 entropy-25-00166-t002:** Correlation coefficients of three pairs of pseudo random sequences.

		Cor
$x_{01}=0.205001025$	$x_{02}=0.205001025000001$	−0.00290
$u_{01}=0.102772828$	$u_{02}=0.205001025000001$	−0.00230
$v_{01}=0.102772828$	$v_{02}=0.205001025000001$	−0.00016

**Table 3 entropy-25-00166-t003:** Information entropy.

${\mathrm{Len}}_{\mathrm{Seq}}$	5000	10000	20000	$10^{5}$	$10^{7}$
$H\left(S_{1}\right)$	7.9904252203	7.9953919349	7.9975813926	7.9994779223	7.9999951973
$H\left(S_{2}\right)$	7.9899963471	7.9949233070	7.9974845070	7.9994868500	7.9999952355
$H\left(S_{3}\right)$	7.9895789685	7.9947618385	7.9970912252	7.9995475969	7.9999954916

**Table 4 entropy-25-00166-t004:** Key sensitivity evaluation based on Bit Change Rate (BCR).

		BCR
$x_{01}=0.105001025$	$x_{02}=0.105001025000001$	49.9994
$u_{01}=0.201772828$	$u_{02}=0.201772828000001$	50.0209
$v_{01}=0.218667888$	$v_{02}=0.218667888000001$	50.0014

**Table 5 entropy-25-00166-t005:** Randomness test by NIST SP800-22 for the PRNG.

*Test Name*	*p-Value*	*Pass Rate*	*Results*
Frequency	0.657933	98/100	Pass
Block Frequency (m = 128)	0.051942	100/100	Pass
Cumulative Sums (Forward)	0.224821	98/100	Pass
Cumulative Sums (Reverse)	0.455937	98/100	Pass
Runs	0.514124	100/100	Pass
Longest Run of Ones	0.262249	100/100	Pass
Rank	0.955835	98/100	Pass
FFT	0.883171	98/100	Pass
Non-Overlapping Templates	0.867692	100/100	Pass
(m = 9, B = 000000001)			
Overlapping Templates (m = 9)	0.574903	100/100	Pass
Universal	0.474986	99/100	Pass
Approximate Entropy (m = 10)	0.574903	100/100	Pass
Random-Excursions (data3)	0.422034	67/67	Pass
Random-Excursions Variant Serial (data5)	0.922036	67/67	Pass
Serial Test 1 (m = 16)	0.883171	99/100	Pass
Serial Test 2 (m = 16)	0.350485	100/100	Pass
Linear complexity (M = 500)	0.275709	99/100	Pass

**Table 6 entropy-25-00166-t006:** Speed comparison.

PRNG	Mean Time Cost
Proposed PRNG	0.3514
Ref. [21]	3.8989
Ref. [40]	108.1568
Chebyshev map	0.8056
Ref. [41]	2.2114
LFSR with discrete chaotic map [42]	3.4530

## Data Availability

The data presented in this study are available on request from the corresponding author.

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
