# Peer review of "A Pseudorandom Number Generator Based on the Chaotic Map and Quantum Random Walks"

_entropy, 2023, doi:10.3390/e25010166_

Round 1

Reviewer 1 Report

Paper is long enough. The investigated problem is covered sufficiently extensively. References also reach the expected number. Basically, it is a bibliography that is up-to-date. Li-Yorke chaos is certainly an interesting phenomenon. The task of the authors is to describe the process of generating pseudorandom numbers. Theoretically and empirically, this is a really detailed paper. This is a significant contribution to the practical application of chaos theory. Similar to deterministic chaos, there are also Lyapun exponents that the authors analyze in this type of chaos. We appreciate the proposal to describe the new chaotic system. The authors analyze the complex behavior of this system. The analysis is very detailed. The only thing I recommend is to slightly extend the conclusion of the paper. After this step, I recommend publishing the paper.

Reviewer 2 Report

The authors are advised to discuss merits or demerits of some chaos-based prng papers in the Sec-1.

I am surprised how the authors justify the computation overheads of square root operation present in their map (eq. 4 or 7 or 15).

The authors have done good performance assessment. But, since the study is solely meant for PRNG design. Hence, it is mandatory to do the exhaustive randomness analysis of the sequences generated from the anticipated method. Hence, authors are advised to add the outcomes of the randomness study from the TESTU01, and DieHard randomness test tools.

Also. presents the rate of generation of bits for real-time applications and compare the results with existing PRNG schemes.

Author Response

Dear reviewer

Thank you for your attention and recommendation, please see the attachment.

Reviewer 3 Report

The paper presents a new PRNG approach based on chaos theory, it is well written, clearly presented and extensively analyzed; the obtained results are convincing, it is probably the most well written proposal of such a PRNG that I have seen. (and I have seen a few)

I have only minor revision suggestions for the authors:

1. The use of the term "key" instead of "seed" and "key space" instead of "seed space" can be annoying at places; please use the terms "seed" and "seed space" instead. 

Also the mention that a PRNG is used in cryptography mainly to generate keys, I find debatable.

2. NIST statistical analysis of the PRNG can and should be improved

Testing ONE single sequence (even a long one) is not enough; several sequences must be tested, some would say at least 100. Now if ALL 100 sequence pass ALL tests, you did something wrong. Hint: alpha =0.01 so some sequences must fail even if the generator is perfect.

Also: NIST STS includes high level tests that aggregate the individual tests, I would very much want to see them included in the paper.

3. Speed performance analysis

"Speed" usually means throughput for the PRNGs or the TRNG, and it is measured in Bytes/second (or multiples (MB/s, etc.))

Please compute and provide results using this classical metric. 

Author Response

(The authors gave the same response as above.)
